# Exploring the Nexus between Displacement and Land Administration: The Case of Rwanda

**Dimo Todorovski [1,\*] and Jossam Potel [2]** 

[1]  Department of Urban and Regional Planning and Geo-Information Management, Faculty for
    Geo-Information Sciences and Earth Observation—ITC, University of Twente, Hengelosestraat 99,
    7514 AE Enschede, The Netherland

[2]  Department of Land Administration and Management, Institute of Applied Sciences,
    155 Ruhengeri, Rwanda; jossam2003@ines.ac.rw

\*   Correspondence: d.todorovski@utwente.nl

**Abstract:** In conflict situations, many people are displaced because of hostility and arms in the area. Displaced people are forced to leave behind their properties, and this in turn interrupts the relationship between people and their land. The emergency period in particular has been identified as a weak point in the humanitarian response to land issues in post-conflict situations. In addition, during this period of response, most post-conflict governments do not prioritize land administration as an emergency issue due to other social, economic, security, and political challenges, which countries face in the immediate aftermath of the conflict. In the longer run, this results in post-conflict illegal land occupation, secondary occupation, numerous disputes and claims over land, and dysfunctional government institutions that legalize these illegal and secondary occupations. This research explores the nexus between displacement and land administration in a post-conflict context. It uses empirical data from fieldwork in Rwanda, and discusses how government interventions in land administration in emergency and early recovery periods of post-conflict situations affect future land administration during the reconstruction phase. The post-conflict Rwandan government envisaged proper land administration as a contributor to sustainable peace and security as it enhances social equity and prevents conflicts. Thus, it embarked on a nationwide systematic land registration program to register land all over the country with the aim of easing land administration practices and reducing successive land-related claims and disputes. However, the program faced many challenges, among which were continuous land claims and disputes. Our research anticipates these continued land claims and disputes are due to how land issues were handled in the emergency and early recovery period of the post-conflict Rwanda, especially during land sharing initiatives and Imidugudu (collective settlement policy).

**Keywords:** displacement; land administration; post-conflict; Rwanda

## 1. Introduction

The most serious concerns stemming from violent conflicts are the number of people killed, widespread destruction of properties, and displacement. Because of the hostilities, people flee for their own safety and have to leave behind their properties. Displacement disrupts the relationship that people have with their land. This disruption has long-lasting or even permanent effects in a customary land tenure setting or even in the formal arena of land administration. The people-to- land relationship is usually represented in a customary or formal land administration system.

When primary displacement occurs during the conflict or post-conflict period, it results in illegal land occupation, secondary occupation of abandoned properties, numerous disputes and contradictory

claims over land, and dysfunctional government institutions that legalize these illegal and secondary occupations. Short term, mid-term, or protracted conflicts have different negative effects, and the situation becomes more complex at the end of the conflict when masses of displaced people return to their places of origin in a short time. When displaced people return, they usually find their properties destroyed, illegally occupied by secondary occupants, or in a totally different situation compared to the one they left behind when they were displaced. This is a serious threat to security and can spark new conflicts.

Land has been identified as one of the weak points in international response capacities [1]. The past decade has been witness to a growing understanding of the vital importance of land issues and of addressing housing, land, and property (HLP) challenges within the context of post-conflict peacebuilding. At the same time, experiences show that there are only a few cases where land issues were appropriately addressed in the post-conflict period. International response organizations in this period mainly focus on the provision of food and shelter for internally displaced persons (IDPs) and refugees and on restoring the situation to how it was before the conflict [2]. Research has shown how these issues are addressed from the perspective of peacebuilding and reconciliation. Rwanda is one of the cases where proper land administration is seen as a contributor to sustainable peace and security, as it enhances social equity and prevents new conflicts [3].

This research explores the nexus between displacement and land administration in a post-conflict context for the case of Rwanda. It aims at increasing the understanding about this nexus and identifies which land interventions occurred in different post-conflict periods in Rwanda. In the second section, the main two concepts—displacement and land administration—are discussed based on the available literature. This is followed by a short description of the case and methodology used for data collection in Section 3. Section 4 presents the empirical data from the fieldwork in Rwanda and a synthesis of the results according to three post-conflict periods. We end by drawing general conclusions about the nexus between displacement and land administration.

## 2. Displacement and Land Administration

In this section, the two concepts of displacement and land administration are elaborated upon the available literature. The discussion focuses on what has been written in the literature with respect to the three post-conflict periods because the latter provides the structure for presenting our results from Rwanda in Section 4.

### 2.1. Displacement

Two basic meanings of displacement can be discerned according to social scientific understandings, namely displacement as (1) forced migration by persecution or violence, or (2) a development-induced displacement referring to people being displaced in the name of projects for economic development [4]. Our research pertains to the first definition.

The Internal Displacement Monitoring Centre (iDMC) also defines the displacement of people as a forced movement of people from their locality or environment and from their occupational activities. It is caused by a number of factors, of which the most important ones include armed conflict, natural disasters, and famine. What is clear is that during this displacement, displaced people leave everything behind to begin a new life in the midst of uncertainty and fear. According to the Norwegian Refugee Council (NRC), causes of displacement are multiple and complex. People may be displaced by armed conflict, generalized violence, and/or human rights violations [5].

"Conflicts are becoming more intractable. They are longer—more than 20 years on average—meaning that the people they displace are spending ever-increasing amounts of time away from their homes and communities. They are more complex, as armed groups compete for control over state institutions, natural resources, and territory. As a result of all types of conflict, 65.6 million people have been forcibly displaced from their homes—the largest number ever recorded, according to

UNHCR. Most—40.3 million—are IDPs, people displaced within their own country. Refugees who have fled to another country make up the next biggest group at 22.5 million" [6].

Displacement caused by conflicts, especially armed conflicts, always results in significant changes to land tenure and land administration, as many people may have been displaced during the conflict. This results in secondary occupation, especially of the land that has been left behind by the displaced persons. Host communities are also directly affected, as they face unexpected increases in the competition on land, especially with implications on the access to land, water, and forests, and associated increased risks of environmental degradation [2].

The normative framework for addressing housing land and property rights in the context of displacement is summarized in the 2007 Principles on Housing and Property Restitution for Refugees and Displaced Persons, known as "Pinheiro Principles". "The Pinheiro Principles provide restitution practitioners, as well as states, the UN, and others agencies, with a consolidated text relating to the legal, policy, procedural, institutional, and technical implementation mechanisms for housing and property restitution" [7]. This document is a compilation of existing rights-based documents in international human rights and humanitarian law. It acknowledges that all displaced persons should be protected regarding their HLP rights—the right(s) that they had to their property should be restored, or if that is not possible, they should be compensated. The Pinheiro Principles make some references to land administration issues as well (Pinheiro Principles: 13, 15, 16, 17, 20, and 21). Here, the twentieth and twenty-first principles address the enforcement of restitution and compensation.

## 2.2. Land Administration in Conflicty and Post-conflict Contexts

There is a tight relationship between humankind and land, which is represented in the form of rights, restrictions, and responsibilities (RRRs) to land. Rights on land can be divided mainly into two groups, statutory and customary rights, and they may be defined in the statutory or common law and by customary traditions or informal uses. Within the statutory or common law (also referred to as the formal system), rights to land and real estate are clearly described in land governance related legislation. Because these rights to land and real estate are established through legal instruments, it can be assumed that these rights are protected and secured for the rights holder. Examples of types of rights to land and real estate include ownership, leasehold, freehold, easements, superficies, and rights to profit. Customary traditions (or customary law) are based on unwritten rules, which find their legitimacy in tradition, and these traditions can be different depending on culture, social aspects, economic, and political factors [8].

Land plays a specific role in conflict and post-conflict contexts. Therefore, it is important to acknowledge how land is administered under these circumstances. Land administration is considered as "the process of determining, recording, and disseminating information about tenure, value, and use of land when implementing land management policies" [9]. Unfortunately, post-conflict situations mostly lead to dysfunctional land administration systems, which are characterized by limited prioritization of land policy, discriminatory land law, and poor institutional and regulatory framework, which in turn allows the grabbing of public and private land by powerful individuals and groups, poor management information systems for updating records, as well as weak state capacity incapable of helping IDPs and refugees [10]. Incomplete, out of date, or contested land records can pose a threat to tenure security and to maintaining newly established peace. The issues about land records in post-conflict situations that require appropriate attention are inadequate land records, fragmented responsibilities for maintenance of land records, lost, stolen, fraudulent, and altered land records, and woman and children's property and inheritance rights [11].

Destruction of infrastructure and properties and displacement have large impacts on land and its administration in conflict and post-conflict contexts [12,13]. These issues become more complex after the end of the conflict when people come back to their places of origin in large numbers and usually find their houses and properties burned, destroyed, or illegally occupied by secondary occupants. This is a critical moment of the post-conflict period, and these problems could spawn disputes over land

and properties or even set in motion new armed conflict [14]. It is evidenced that in the post-conflict context, the following land-related problems occur: landlessness, lack of access to land, non-functional land administration systems, forced land transactions, emergency occupations of land, and disputed housing and property rights [15]. "Disputes and claims over land are a very frequent problem in post-conflict settings. Therefore, land dispute resolution mechanisms are viewed as a conditional tool for a good peace process" [16].

Housing, property rights, and land administration are always negatively affected by conflicts, and if not addressed properly in a post-conflict context, they could be reason for new disputes or cause for renewed armed conflict [17]. Looking back at the history of peace agreements reveals that these agreements contain only limited references to land issues. Peace agreements mainly focus on the basic human right that the displaced people should be able to return to their properties with dignity. Only in the cases of Kosovo and Timor-Leste were specific land management and land administration activities integrated into the peace agreement and UN operations. As seen in practice and literature, there is a clear need for putting land issues on the agenda of the international community and to address these issues explicitly in peace agreement documents or national land policies of the states as they emerge from conflict [18].

In the aftermath of the conflict, in many cases, state institutions are not functioning normally, and the state administration is not effectively practicing its core duties. Addressing and solving land disputes and claims in such situations are important but difficult because of other priorities [5]. In order to appropriately tackle land and property problems in post-conflict contexts, it is necessary to be aware of the fragility of the particular context of a post-conflict country and address land issues from the specific perspective of peacebuilding and development [19]. Addressing land and land administration in such contexts should therefore be coherent with overall state building efforts. Takeuchi et al. [13] argue that tackling land and property problems has important implications for post-conflict state building in general, particularly for legitimacy building, because regulating land and property rights via an appropriate land administration is one of the fundamental functions of the state.

*2.3. Post-Conflict Periods*

There is diverse literature on both displacement and land administration in different time periods that focus on the time periods after a conflict. Here, we focus on the three post-conflict periods, namely the period of emergency, the early recovery period, and the reconstruction period. Some authors define these periods according to fixed times, e.g., emergency as lasting one year, early recovery as the time span covering one to three years after conflict, and the reconstruction period as four to ten years after conflict [20]. Alternatively, other authors define different post-conflict periods based on specific activities undertaken in each period [21]. In these latter understandings, the periods should not be understood as absolute, fixed, time-bound, or as having clear boundaries of transition. Similarly, different geographic, ethnic, linguistic, religious, or regional groups may experience different post-conflict periods at different times within a country [22]. Within this research, we focus on the activities undertaken in a post-conflict situation in order to further identify further interventions and to analyze what is functioning and what is not during a specific period. For this purpose, we provide descriptions of the three post-conflict periods in the following sub-sections according to existing research and literature.

2.3.1. Emergency Period

Activities in the emergency period focus on establishing basic governance mechanisms and providing humanitarian services. They are usually undertaken in the immediate aftermath of conflict and before full-scale mobilization of aid and resources has started. During this period, there is often little or no operational governance or rule of law, and extensive destruction of infrastructure occurs and food security is low. These activities should be identified as short-term actions that can be implemented relatively quickly [21]. In many cases, according to literature and practice,

conventional land interventions in this period are not possible. However, the international community has acknowledged that land tools and approaches developed in this period can serve as a basis for the anticipated early recovery and reconstruction periods [23].

### 2.3.2. Early Recovery Period

The early recovery period marks the transitional phase of a post-conflict country during which legitimate local capacities emerge and should be supported with particular attention paid to restarting the economy. This includes physical reconstruction, ensuring functional structures for governance and judicial process, and laying the foundations for provisions of basic social welfare, such as education and health care to foster social stability [24]. In this period, activities for the development of policy include the identification of policy priorities, the establishment of government institutions and administrative infrastructures, recruitment of core experts, development of interim policy and legal frameworks, and the development of long-term policies. It is also a starting period of activities undertaken to implement policy, including activities such as the definition of institutions within the laws, implementation of policy strategies, and policies to support changes in livelihood strategies [25].

### 2.3.3. Reconstruction Period

In the reconstruction period, the rebuilding of society and physical infrastructures continues. A government in a post-conflict situation requires revenues, and land tax can be an important revenue stream. For this purpose, land in a country needs to be well administered [26]. In the reconstruction period, much attention is focused on the implementation of the policies and law enforcement. In land policy, implementation of previously conceived policies on the access of land should be supported through appropriate land legislation, adjudication procedures for land claims and disputes, existing land administration systems, housing strategies, eviction procedures, administration of state-owned land, and administrative structures to regulate private abandoned land and foster transparency [21,22].

Therefore, when exploring the nexus between displacement and land administration, the activities and aims outlined above need to be explored within the structure of the above-mentioned post-conflict periods. In the following section, the methodology on which this research is based is presented.

## 3. Methodology: Selection of the Study Area and Data Collection

The Eastern province of Rwanda, namely the districts of Kayonza and Ngoma, were chosen as study areas because the province's location is one of the places in the country that received a massive number of returnees from Uganda, Tanzania, and Burundi. Many of the returnees from these three countries settled in the Eastern province. These returnees needed land and shelter, which led to various land interventions and land sharing. During the fieldwork, the district officials were consulted on the access to land of the returnees per sectors and population. Consequently, the following two sectors within the province were chosen for a closer investigation—Mukarange sector in Kayonza and Remera sector in Ngoma district. In these sectors, fieldwork activities were carried out for the data collection.

A stratified random sampling method was used to choose respondents for four target groups: the 1959 refugees, the 1994 refugees, the IDPs, and the survivors of the Tutsi genocide. The data collected provide insight into how land issues were handled in post-conflict Rwanda, especially in the emergency and early recovery periods, which later affected land administration in the reconstruction period.

### *Data Collection*

Our research is based both on the literature review and on primary and secondary sources. The latter were collected during fieldwork. Primary data were collected through interviews and focus group discussions as well as field observations. Secondary data were collected from governmental organizations at sector levels, mediation committees, the district and the office of the ombudsman, the

Rwanda Natural Resource Authority (RNRA)—recently renamed into Rwanda Land Management and Use Authority, and national secretariat for mediation committees.

In order to understand the nexus between displacement and land administration in the aftermath of the conflict, the post displacement era was divided into three post-conflict periods, as elaborated in Section 2 of this paper. The empirical part of this research aimed at collecting information based on these three periods.

During the fieldwork, interviews were conducted with 30 respondents from each of the four targeted groups. The focus group discussion involved 12 former leaders representing each of the four targeted groups. The focus group discussions were executed in two different sessions with six participants each, whereby the discussion focused on how they handled land claims and disputes of returnees in the post-conflict period. As this research intended to explore the nexus between displacement and land administration, questions discussed during the focus groups related to how people repossessed their land in the post-conflict era, the level of secondary occupation, land grabbing by the elites and government officials during the emergency and early recovery periods, and the effects on land administration in post-conflict periods.

Question guides for interviews with displaced persons and for interviews with members of the administrative group (mayors, land officials, and former state officials) were developed pre-fieldwork. All interviews were transcribed in the form of a fieldwork report, and a summary of minutes from the focus group discussions was prepared. Details about this process can be found in Chapter 3 and appendixes 2–5 of the research undertaken for the Master's thesis project [3].

## 4. Results and Discussions

### 4.1. Conflict and Post-Conflict Rwanda

The conflict in Rwanda is considered a protracted conflict that took place during the period between 1959 and 1994. Land issues are considered a major cause in that they were used as a fueling factor for the increase of ethnic tensions [27]. Political representation and unresolved governance issues are additional causes [28]. The first violent conflict in 1959-1963 in Rwanda resulted in half a million refugees crossing the borders to neighboring countries. This group of refugees stayed in refuge until 1994. For the period after the first violent conflict, the government of Rwanda used land as a political tool, redistributing abandoned properties to their political followers and military officers, causing illegal occupation by secondary occupants. The second violent conflict in Rwanda finished in 1994 with a genocide that caused more than one million people to die, a large number of IDPs, and more than two and a half million refugees [29]. The Arusha Peace Agreement Document was signed in 1993. Only after the victory of the Rwanda Patriotic Front (RPF), which took control of the whole country by mid-July 1994, did the security situation improve. Displaced people began to return. In process terms, this moment can be considered as the start of the post-conflict period.

The emergency period in post-conflict Rwanda lasted from 1994 to 1997 and was marked by unity and reconciliation activities. After the humanitarian crisis, the main focus was on the provision of shelter, food, and providing basic living conditions for the population returning in masses [30]. Actors on the ground did not seem to understand the importance of land issues at this time; they were totally out of the focus of activities [31]. The time from 1997 until the end of 2002 can be seen as the early recovery period. This period involved the development of a legal framework, national policy developments, setting up new government structures, and development of strategies for implementation. The period after 2003 can be identified as the reconstruction period. It involved the implementation and execution of the legal frameworks, national policy, and development programs [25].

In the following sub-sections, the empirical data about displacement and land administration in post-conflict Rwanda are categorized into the above three post-conflict periods (emergency, early

recovery, and reconstruction periods) in order to elaborate on the nexus between displacement and land administration.

*4.2. Displacement and Return during the Emergency Period in Post-conflict Rwanda*

In the aftermath of the genocide of the Tutsi, many "1959 refugees" who were living in neighboring countries such as Uganda, Tanzania, Democratic Republic of Congo, and Burundi returned back to Rwanda in big numbers in 1994. At the same time, the so-called "1994 refugees", mainly the Hutus who fled Rwanda in fear of retribution, took refuge in the same counties mentioned above. This left the land (prior to 1959 occupied by Tutsi and given to Hutu after 1963) vacant to be occupied by the returning Tutsi, the "1959 refugees". Due to the impacts of the war and the genocide, many of the properties, including houses, had been destroyed. The authorities decided to locate the returning Tutsi to houses and fields that had been abandoned by the Hutu. During 1996-1997, the "1994 refugees" returned and witnessed that the land they had been occupying for over three decades was now occupied by other people who also claimed legal rights to the same land. At that moment, the country was facing the challenge of two successive legitimate and yet competing claims from two groups of returnees. One former local leader explained that "the government authorities convinced them to share".

Although some respondents during the fieldwork interviews did mention that local leaders at the cell and sector level were in charge of land issues, many respondents agreed that by that time, nobody was seriously considering how to handle the land issues. Everybody was concerned about the safety of returning refugees and how to provide them with food and shelter. There were no land related complaints at the time. The interview with the former mayor recounts that "by that time, we were not thinking and working on the land issues. What mattered for us as authority was to provide food and shelter for returning refugees. Some individuals used that period of emergency and went on to grab land that was left vacant by 1994 refugees" [3]. This moment of the emergency period shows that where displacement had occurred, land administration could have been prioritized. Elite groups normally take advantage of dysfunctional government to grab and legalize the land that either belonged to the displaced people or to the public.

From the interviews with local people from the three groups, the "1959 refugees", the "1994 refugees", and the IDPs, the data reveal that a big number of people never repossessed their land, and even for those who did repossess it, it was not of the same extent as it was prior to displacement. This shows how displacement affects land tenure and land administration with long-term and permanent effects. The Table 1 below shows the synthesis of the data analysis of the repossession of land in post-conflict Rwanda.

For better visualization, the same synthesis of the data analysis on the repossession of land in post-conflict Rwanda is also represented in the following pie chart in Figure 1.

Discussion:

In the case of Rwanda, there was no fully operating public professional body in charge of land administration during the emergency period. Land issues tended to be handled informally, sporadically, and by officials with little skills and knowledge in land administration. This resulted in the public undertaking either illegal or informal actions, either intentionally or due to ignorance. Thus, the emergency period was characterized by responding to emergency security challenges, social and political challenges related to the big number of returning refugees, and how they could best get food and shelter. In this period, land-related issues and challenges were not considered priority issues.

**Table 1.** Fieldwork results on land owned by displaced persons [3].

| No | Assessment tool | Responses | Frequencies | % |
|----|-----------------|-----------|-------------|---|
| 1 | Did you occupy land before the conflict | Yes | 31 | 100% |
| | | No | 0 | 0% |
| 2 | Was your land occupied by others when he/she returned | Yes | 29 | 94% |
| | | No | 2 | 06% |
| 3 | Who had occupied your land | 1959 Returnees | 9 | 31% |
| | | 1994 Returnees | 2 | 7% |
| | | Internally displaced persons (IDPs) and other people | 13 | 45% |
| | | State | 3 | 10% |
| | | Church | 2 | 7% |
| 4 | Did you reoccupy your original land | Yes | 18 | 62% |
| | | No | 11 | 38% |
| 5 | Reoccupied the entire parcel | Yes | 3 | 17% |
| | | No | 15 | 83% |
| 6 | How long did it took to reoccupy it | Below a month | 14 | 78% |
| | | Above a month | 4 | 22% |
| 7 | Did you possess any evidence | Yes | 0 | 0% |
| | | No | 31 | 100% |
| 8 | How did you prove | Witnesses | 9 | 29% |
| | | No proof needed | 22 | 71% |

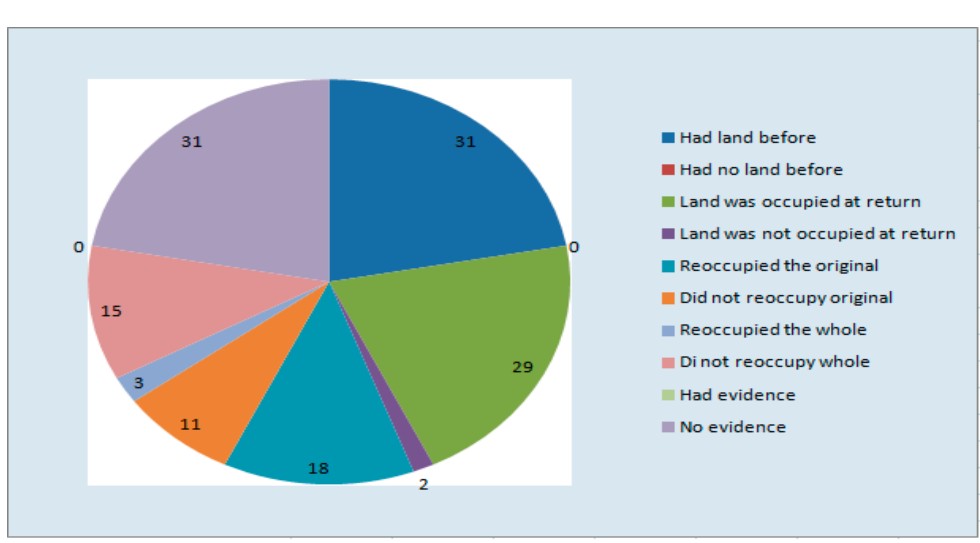

**Figure 1.** Repossession of land in post-conflict Rwanda.

*4.3. Early Recovery Period in the Case of Post-Conflict Rwanda*

The early recovery period was a bit different from the emergency period. In the early recovery period, the security situation started to stabilize. Since people started to plan for the future, they needed land to cultivate crops and housing, and here land claims and disputes started to come to the fore. The data collected reveal how the people and Rwandan authorities reacted to land claims and disputes in the early recovery period of post-conflict Rwanda.

During the interview with the director of the One Stop Center of the study district, he responded that "the authority could not address the issue before it could rise, as, by that time, many people were suspicious of each other so they had to wait and seek guidance or the regulations from the authority".

After repatriation of the refugees from both the 1959 and the 1994 groups, the country had entered an early recovery period of a post-conflict state, life had turned to normal, and government institutions started working. However, land issues also grew in complexity. The fieldwork researcher wanted to identify how the authority handled the land issue after the repatriation of the displaced persons, and from respondents' answers, it was found that all confirmed the complexity of the situation. This is how an interviewed respondent from the One Stop Center described it: "It was a challenging moment, with no legal direction to solve different overlapping rights to land by different people from different groups of returnees, so we gave local people chances to search for a solution themselves".

Complexity in land matters manifested, for instance, in parcels of land with different claimants. These claimants varied between two to four, all claiming rights to ownership, many of whom had genuine reasons. Thus, there was no other option but to let them all share the same parcel of land, as explained by one respondent and a former leader: "We had no clear and legal formula to solve the claims of these people, so we gathered local leaders, and in a period of eight days we had the task to find the solution. The outcome was 'land sharing' and Imidugudu settlements, because these came out as the only approach that would allow us to cope with the situation of solving a land issue and at the same time could be used to exemplify political achievement during national campaigns for peace, unity and reconciliation". Land sharing policy was an informal policy and was implemented differently in different sectors depending on the prevailing conditions per each sector. In principle, the "1959 refugees" had to share the same parcel of land with the "1994 refugees", further adding to the complexity and precariousness of the situation due to the fact that these two groups knew each other as members from the periods of violent conflict. The basic framework for setting land contestations was provided in the Arusha Accord (top-down approach), but the implementation of land sharing and Imidugudu settlements was bottom-up [17].

The Arusha Peace Agreement Document (PAD) recommended that, in relation to land matters, in order to promote social harmony and national reconciliation, refugees who left the country more than 10 years ago should not reclaim their original properties because the land might have been occupied by other people [32].

According to the peace agreement between the Rwandan Government and the RPF, the "1959 refugees" were not allowed to reclaim and repossess their land. They were instructed to be resettled to state land provided by the government. The so-called "Imidugudu" program was meant for the returnees that did not manage to get land via land sharing. They were offered to take state land in national parks [25]. This was not welcomed by all "1959 refugees" and often resulted in their return to their previously possessed land before the displacement took place in Rwanda. In the early recovery period, the "1959 refugees" legally challenged the government over the right to repossess their land, to which the government finally had to agree in order to avoid new claims and disputes. For instance, Mr. Ivan[1], an interviewed returnee, wondered, "when you look to the way these people were expelled, you would see that it would be unfair not to allow them to resettle in their place of origin". Thus, the authorities decided to let these people reoccupy their land and share with the others that had acquired it in the period 1962–1994 [3].

As seen above, secondary occupants could be 1959 returnees, 1994 returnees, IDPs, state, or the church. Since it was confirmed by the data collected that the majority of "1959 refugees" did not reoccupy their land, it was paramount to know how the Rwandan government solved such an outstanding legitimate claim. The interview with the former minister of local government answered the researchers' concerns: "The law was clear, all the land belonged to the state. So the government took

---

[1]    Not real names.

control of land, it was easy to declare the state requisition since the land tenure allowed it. That's why the sharing principle was deployed and people had to accept it because they had no alternative. As leaders we had no other legal solutions; so we had to opt for unconventional ad-hoc solutions".

Such as in the emergency period, at the beginning of the early recovery period, government had limited staff with required skills and knowledge to deal professionally with land issues. In post-conflict Rwanda, the actors involved in implementing the land sharing process were local people who were tasked to implement it themselves while supervised by their local leaders. This was the moment when the land claims and disputes over land were at their peak. To conduct the land sharing process, a land claim committee of four people was chosen, representing the four groups, respectively—that is, one representing the "1959 refugees", one the "1994 refugees", one the IDPs, and one for the survivors of the Tutsi genocide. All were supervised by the cell chairman. Each cell reported to the sector councilor. One interviewed respondent answered that, "As we were still in a period of mistrust and suspicion, we choose committees from different groups, that is one person from 1994, one from 1959, one from genocide survivors, one from IDPs and the head of the cell as the representative of the authority" [3].

However, in addition to the many land claim controversies between the two main groups of the displaced people during the early recovery period, other land claims arose, adding to the nature and the degree of the complexity. Fieldwork data collection revealed new challenges that surfaced, emanating from practices of land sharing.

The following excerpt provides an example that illustrates the complexities involved in the case of shared family land:

> *Jean and Pierre are brothers; and together they owned 3 hectares of land before 1994. In 1994, Pierre fled to take refuge in fear of retribution. Immediately after the end of the war, Jean had to share the land with another person, Mark. Mark was a 1959 returnee. Pierre returned in 2000 and asked Jean for his part. But Jean no longer had Pierre's part. So, Jean asked Pierre to share Jean's part of the land that the latter had received during the land sharing process. Pierre did not accept this and lodged a land claim through the mediation committee for restitution of their family land against Mark. Pierre lost the case and later decided to turn to an ordinary court against the state.* [3]

Discussion:

The early recovery period in post-conflict Rwanda was characterized by an improved security situation. Once all returnees were taken care of, people and Rwandan authorities started planning their future through appropriate reforms, the new constitution, and established appropriate legal frameworks. Because in this period all displaced people returned, land issues became a paramount challenge. Tackling the complex land issues was achieved through land sharing and Imidugudu settlements, which were ad-hoc solutions that intended to support peace unity and reconciliation. With respect to land issues, this period witnessed a peak in land claims and land disputes. The setting up of the mediation committees at village/settlement levels, including members of all four affected groups, showed to be a very helpful and effective tool in addition to the three level state land dispute mechanisms. Through instruments like the PAD, these measures were enforced to pave the way forward towards peace, unity, and reconciliation.

*4.4. Reconstruction Period in the Case of Post-Conflict Rwanda*

The period of reconstruction begins when a country is stable and embarks on institutionalizing long term development programs to foster economic growth and sustainability. In Rwanda, one of the development programs was the Land Tenure Regularization (LTR) program from 2008 to 2014. LTR aimed to improve agricultural productivity, enhance economic development, and reduce social tensions. The program was in line with the adoption of the new constitution, national land policy, promulgation of organic land law (currently changed to new land law), and the establishment of a public institution with the task to oversee the management and use of land in the country.

While during this period, the implementation of land reforms determined by land policy and law was meant to foster economic growth, authorities continued to face the challenge of an increasing number of contradictory land claims and disputes. These originated in the emergency and early recovery periods and how land issues had been handled during these earlier post-conflict periods. While many of these land claim and distribution issues were family matters, they did often have their roots in the practices of land sharing promoted in previous post-conflict periods. The words of one respondent summarize the situation as follows: "Almost all of the claims are family cases but mostly related to land sharing". The case of Mr. Green illustrates the complexity of the situation:

*Mr. Green left the country in 1959 after the massacre of his father and grandfather, They had 30 hectors of land with a forest and a house in it. His land was redistributed to local people by the then government in 1962. Mr. Green returned back in 1994 and found that those who had been occupying the land were displaced just before he returned. Mr. Green re-occupied his land. During the 1997 land sharing, he refused to share the land with anybody, claiming that the land used to be his family's land and of all the descendants of his late grandfather the late Mr. Gray. As the country regained stability, many of the people who had received the land from the then government and occupied that land (in the period 1962-1994) returned and started to lodge claims to the authority over the right to the same land that Mr. Green had re-occupied. In 2009, there were 37 claimants all claiming rights over the same 30 hectares of land.*

Throughout the research and interviews conducted in [3], the responses from different interviewees revealed that many of the complaints received were related to: (1) the Tutsi-Hutu conflicts of 1959 and 1994; (2) the land sharing practices of 1996–2003; (3) land of the genocide survivors, especially orphans' land, which was now occupied by others or by the state; (4) conflicts between family members due to how land sharing was handled; and (5) related to land taken by the Imidugudu settlement.

Exploring the nature of these competing claims more closely, it was found that some of the claims made by one of the partners in land sharing were fraudulent. This resulted in only partial official registration of documents and family members filing claims against one another, contesting the results of land sharing. In some cases, land sharing was unfair. In other cases, some family members used the situation of other family members' longer time absence. Complaints related to the Imidugudu settlement concerned mainly land that belonged to the orphans of the 1994 genocide. From the nature of these complaints that were further processed in the official state legal system (the three level land dispute mechanism), it became clear that the continued contradictions and new filing of land claims, as well as land related disputes, were associated with previous periods of displacement.

Although some interviewed respondents attributed contradictory land claims and disputes to poor governance that led to the displacement of people, many respondents attributed the continuation of the problem to the post-conflict period and how land issues had then been handled. As one respondent stated, "We receive claims that are not related entirely on displacement, but on how the land issue was handled after the displacement, especially when land sharing occurred". When asked how by the researcher, the response was that, "Some claimants say that, for example, family land was given to one person who shared with a returnee without other family members getting anything". Based on secondary data received from the National Secretariat for Mediation Committees, it was observed that many of the cases handled by these committees are land related, and given the responses received from respondents, many of the cases received are related to displacement. The following Figure 2 represents the variety of cases submitted to the mediation committee in one cell; here, land-related cases dominate, making up 41% of the nine different types of cases.

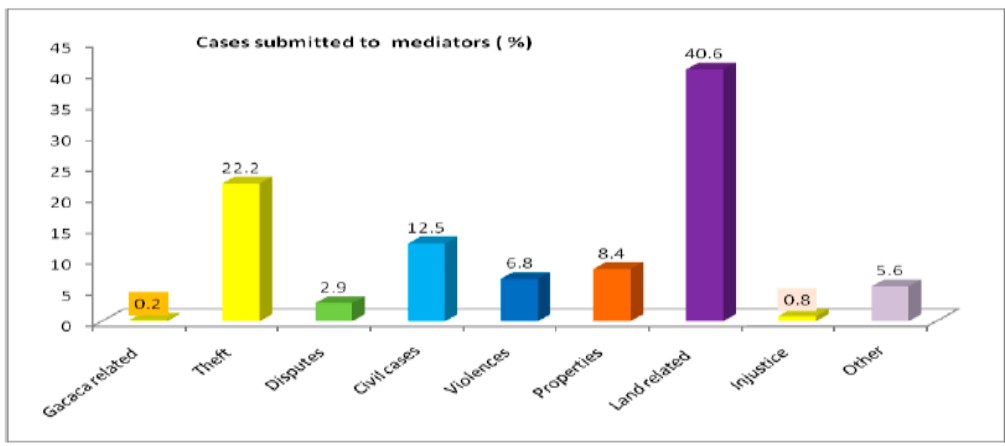

**Figure 2.** Variety of cases submitted to mediators committee in a cell 2012 [3].

In addition to the findings from Figure 2, respondents from National Secretariat for Mediation Committees acknowledged that all received land related cases were solved before or during the implementation of the LTR program. The received and solved land related cases by these committees coupled with almost all land sharing cases (for which "the government authorities convinced them to share", as stated in Section 4.2) and disputed family matters related to land suggest that land disputes and claims were in big numbers and serious threats to the fragile post-conflict situation.

Reviewing the general report of the LTR program (secondary data source from fieldwork from RNRA) for the whole territory of Rwanda, it was noted that 0.13% of demarcated parcels with complete information were under dispute in court and therefore remained unregistered as per LTR regulations. In addition, there were 1,494,943 non-registered parcels (14% of demarcated land), the information for which was incomplete due to the fact that the true owners had either been killed and no close kin were found to take over the land, or the original owners had been displaced and never returned. The following Table 2 provides more detailed figures from the general report of LTR program.

**Table 2.** Showing general report of Land Tenure Regularization (LTR) on disputed and unregistered land [3].

|  | Demarcated | Complete Information | Incomplete Info | Disputes in Court |
|---|---|---|---|---|
| General Total | 10,480,871 | 8,866,199 | 1,494,973 | 11,809 |
| Percentages |  |  | 14% | 0.13 % |

Land parcels that were under scrutiny regarding claim legitimacy or under dispute during the LTR process were skipped in the process and registered in a claims register, and the respective claimants were directed or referred to competent authorities or the court. Once a decision was taken for one of such parcels, it would then be registered again under the final, true owner's identity. An interviewee summarized the process as follows: "During LTR, we used a claims register, where we registered all the parcels that were in conflict, of which no owner of that land would be registered on it until an administrative document is produced by the party that could prove to be the true owner or by court decision".

The case study of Gillingham and Bruckle (2014) on LTR identified the following as key success factors for the LTR process: political commitment, a detailed LTR approach developed in an earlier phase of the program, and the program's flexibility. Their case study identified some points that need attention in the longer term—further development of the land administration system, as well as financial and judicial sustainability are required [33].

Discussion:

It was not until the reconstruction period that the government of Rwanda really recognized the importance of land and land administration as important state functions with high priority for the government's political agenda. The LTR program and appropriate legal institutional and organizational adjustments were identified. Ad-hoc policies and activities performed in regard to land sharing and Imudugudu settlements that were not conventional but rather aimed at the development of peace, unity, and reconciliation were developed further, and the land was registered during the LTR project. Nevertheless, the continued high number of land claims and land disputes directly related to concerns about longer term land tenure security hampered these efforts.

Because claim competition and disputes during this period originated from previous post-conflict periods, the example illustrates the importance of bringing land formalization and administration to the forefront of post-conflict activities early on in the peace building and state development process.

## 5. Conclusions

The objective of this research was to explore the nexus between displacement and land administration in post-conflict contexts, drawing on the case of Rwanda. The study was based on a literature review and fieldwork for primary and secondary data collection. A synthesis of the main results as well as brief discussions were presented according to the three post-conflict periods identified from the literature.

Our study revealed that during the emergency period, the focus of activities was on establishing security, tackling social and political challenges related to the big number of returning displaced refugees and IDPs, and how to provide food and shelter. Land formalization and administration did not receive priority during the emergency period. In the early recovery period, when most of the displaced persons returned, the Rwandan government turned its attention to the complexity of the land situation. Land related issues were tackled with ad-hoc land policies such as land sharing and the Imidugudu settlement rather than being based on the Pinheiro Principles. A positive aspect during this period was that land issues were addressed (as in the Arusha Agreement) in the context of peace, unity, and reconciliation by putting in place mediation committees and land dispute mechanisms. Another early recovery characteristic was a peak in land claims and land disputes, all related to previous displacement dynamics. All land challenges received appropriate political attention during the reconstruction period. The Rwandan government adopted the LTR program, during which the effects from previous ad-hoc policies and land interventions were addressed and parcels officially registered. LTR still witnessed a large number of land claims and disputes, which could negatively impact land tenure security and result in new conflicts.

From our research, we conclude that the understanding of and governmental focus on the relationship between displacement and land administration interventions are essential for post-conflict peacebuilding, and this applies to all three post-conflict periods. Specifically, our study indicates that issues pertaining to land distribution and related disputes need to be included early on in the post-conflict era, and other cases may be served well by including land related interventions in the PAD.

Rwanda is a case where displacement, ad-hoc land policies, (contested) land claims, and land disputes occurred. All of these gradually evolved into a formal land administrative approach. In other words, at least in the case of Rwanda, the nexus between displacement and land administration is perhaps best described as an ongoing process across post-conflict periods. The aim of developing an appropriate and functional land administration is therefore an essential contributor for sustainable peacebuilding and security—also because it enhances social equity and prevents new conflicts.

**Author Contributions:** Conceptualization: J.P. and D.T.; formal analyses, investigation, data collection and writing draft: J.P.; writing, review, editing, supervision: D.T.

**Funding:** This research received no external funding.

**Conflicts of Interest:** The authors declare no conflict of interest.

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
