# Peer review of "Exploring the Nexus between Displacement and Land Administration: The Case of Rwanda"

_land, doi:10.3390/land8040055_

Round 1

Reviewer 1 Report

The case of Rwanda is a very interesting case to study the responses of the state to land issues deriving from displacement and return. The merit of this paper is that it discusses this case on the basis of original fieldwork. Also the periodisation of land issues in emergency, early recovery and reconstruction is useful and original.

The analysis as presented, however, is not always easy to follow and shows some gaps. Here I list the major issues that I consider require attention:

- Problem statement/central research question: the paper refers repeatedly to the idea that it wishes to explore the nexus between displacement and land administration, but it does not make clear what exactly are the questions about this nexus that are of interest to the authors. The introduction should provide a clearer definition of the knowledge gap that the paper aims to address.

- Provide more contextual information on Rwanda: the paper is very brief on the (extended) period of conflict that is being considered and does not introduce some of the central terms (such as Arusha Accords). Some more background on why what population groups fled, in what numbers and for what periods of time, would help to situate the arguments around the land problems.

- More information on the land sharing arrangements in the 'early recovery period': what exactly is land sharing, and especially, who was enforcing this. Was it bottom up? Was it government policy? Was it meant to be a temporary or lasting solution? More sources should be cited on these land sharing agreements. The term imidugu is not duely explained and it remains unclear how it related to displacement, land claims and land administration.

- The analysis in the third period (reconstruction), requires elaboration. The LTR (or LRT?) policy requires more introduction. What was the political agenda it embodied? What elements did it contain? How was it implemented? What was the role of international donors? The table on land issues (Fig 2) requires more explanation: how central  was the role of these mediators? Did they respond to the state? The term gacaca is not explained.

- The material and quotes on how land sharing led to clashes within families is very interesting. I think it is these problems that the paper refers to when it says - in several places- that the problem was 'how land issues were handled'. This statement would merit  more elaboration: what exactly went wrong, according to the people interviewed, in this 'handling' and who was responsible for this 'handling'? It remains unclear whether the paper aims to put the 'blame' on the individuals who opted for land sharing or on the government that had been suggesting land sharing as an option.

- The central claim voiced towards the end of the paper, that the land administration policy in Rwanda in the post conflict reconstruction phase is an example for other cases, remains unsubstantiated. To substantiate it, the paper would need to explain in what ways it sees the policy as successful and how it understands that this success was produced. The claim relies on own research, but would need to cite broader literature on land policy in Rwanda (I would expect some of that literature to be more critical and this piece would need to engage so some degree with these criticisms). The paper would need to explain more clearly how the land policy solved the 'mishandling' identified in an earlier phase.

All in all, this is a paper with significant potential but that needs to make its arguments with more precision.

Author Response

Dear reviewer, Many thanks for your detailed review and constructive comments/remarks for improvement. Much appreciated!

- Problem statement/central research question: As per your suggestion, knowledge gap is sharper and clearly delineated now in the Abstract and Introduction sections. 

- Provide more contextual information on Rwanda: This is improved by including an additional sub-section about the “Conflict and post-conflict Rwanda” in Section four - results. More contextual information on Rwanda, Arusha Peace Agreement, and numbers about the displaced and returned people in different time periods (sections 4.1 and 4.2) 

- The analysis in the third period (reconstruction), requires elaboration. This section is improved by better introduction and explanation of the political situation in regard LTR program. Figure no. 2 is additionally described.

The material and quotes on how land sharing led to clashes within families is very interesting.... We are on the opinion that handling land issues from perspective of peace reconciliation and social harmony (as in Arusha Accord) served the purpose of peacebuilding. However if those land issues were handled earlier, appropriately addressed in the PAD long-term effects would be more positive and beneficial for displaced. However section 4.4 is improved with Edited English and better reading flow.

- The central claim voiced towards the end of the paper,...This is improved with updating parts of section 4.4 and sharper Conclusions. Including new more critical literature is taken into a consideration, however, this paper has already 8000 words (which is the maximum word limit).

Reviewer 2 Report

I have read the manuscript entitled, Exploring the Nexus between Displacement and Land Administration; the case of Rwanda. My understanding of the study presented in the manuscript is that it strives to fill the gap that exists in post-conflict land administration studies. The focus is on the interface between conflict displacement and land administration. I consider the paper relevant for publication pending the following improvements:

Title: I find the title appropriate but would wish that the authors replace the “semicolon (;)” in the title with a colon (:).”  A colon (:) is definitive for expressing the title, and so, will be more generally appropriate for paper captioning and titling.

Language: The language (diction) of the paper is understandable, but some expressions can come up as misleading. For instance Lines 208-211 seem not to carry the message across appropriately (even though the sentences are correct. In that section (3.1), the authors are meant to be reporting the how-to aspect of their study, but it appears that the sentences are mere assertions about how the research is done. There are many such sentences in the manuscript which needs fixing. 

Theory: The author(s) say, “The two concepts: displacement and land administration are deeper elaborated in the rest of this chapter based on the available literature.” Lines 66-67). However, the paper did not fully achieve this. One question the author(s) could ask is: can a reader grasp land administration merely by reading this article? In my opinion, this was not deeply treated. The paper lacks a well-founded theory grounding to enable readers to grasp it beyond its reporting style. There is a little bit of scholarly that one misses in section 2 which appears more like a definitional chapter and description of what displacement and land administration entails. Some work needs to be done. For instance, introduce strong literature on “development and conflict theory” (or discourse in that line). It will enable readers to understand (later on in the paper) the nexus in its broader sense -that is, land administration as a possible tool for “peacebuilding” in the displacement (post-development) perspective. 

Methodology: The author(s) say: “Interviews are conducted with 30 respondents from each of the four targeted groups. The Focus group discussion involved 12 former leaders, representing each of the four targeted groups as well. The focus group discussion was executed in two different sessions with each 6 participants, whereby the discussion focused on how they handled land claims and disputes of returnees in the post-conflict period” (lines 223-227). How were the interviews administered, recorded, interpreted and then analyzed? What were the questions (at least, mention the most relevant ones)? Of what use was the Focus Group Discussion? How long did these interviews take on the average? Why interview 30 people? An interview with 10 persons lasting about 45 minutes would have gulped 450 minutes (more than 7 hrs.)  of airtime, and if recorded would cost some big bites of storage space. How do you analyze this? Then think of 30 persons. My point is that we need clarity in terms of detailed procedures in the methodology.  

Result: The paper discussed “Displacement and land administration in conflict and post-conflict Rwanda” (lines 232-454), and then concludes. I was left thinking that something was left out. Most of what has been done in this section was to present a scenario of events or situation of important issues with some suggestion from learning points. Maybe another specific section (before the conclusion) could have helped to strengthen the paper in regards to making a core contribution to post-conflict land administration studies. I miss a specific framework presented to enhance the identified nexus towards peacebuilding, stable land administration or… a core idea going forward.

Referencing: The current state of the reference section is acceptable, but the author(s) need to format the reference list into the “Land” format. They should follow the instruction to authors. All the references listed (from number 1 to 31) are presented in the wrong format. 

Author Response

Dear reviewer, Many thanks for your detailed review and constructive comments/remarks for improvement. Much appreciated!

Title: This is taken into consideration and improvement made as per your suggestion.

Language: English language and the reading flow was improved with assistance of two English native speaking persons.

Theory: This section was improved by reorganizing the content and adding one more paragraph about the land administration sub-section.

In continuation a use of literature about ‘development and conflict theory’ is adequately inserted in the sense that development aid aims in same objectives as peacebuilding interventions 

Methodology: The Methodology section initially is improved by a new title reflecting more in details what the reader can expect in this section. It includes sharper selection of the study area, data collection part with mentioning several questions asked in the interviews and link to the full questionnaires for the interviews and the focused group discussions. It is improved by explaining in details how the data was collected.

Result: This section is initially improved by renaming it into: Results and discussions (as suggested by reviewer no.3). It has additional sub-section about the “conflict and post-conflict Rwanda”, and the following three subsections end with a brief discussion. Additional section only discussion of the results was taken into consideration, but the length of the paper would exceed the limit of 8000words given by the editors.

Referencing: This is taken into consideration and the references are improved according to the LAND guidelines.

Reviewer 3 Report

This paper addresses an important issue of land administration in Rwanda. There are few issues that the authors need to address or include in the paper to enhance its scientific quality.

Introduction

First, I  would suggest that the authors delineate the knowledge gap in the literature and how they tend to contribute to narrowing such gap in the introduction. In lines, 58-64, the authors were trying to outline the organisation of the paper, which is ok but I would like them to change the word "chapters" to "sections".

Methodology

In the methodology only the data collection tools were described but the data collected were analysed was not included. I suggest that the authors outline how the data was collected. 

Results and discussion

I would suggest that the authors make section four results and discussion.

Author Response

This paper addresses an important issue of land administration in Rwanda. There are a few issues that the authors need to address or include in the paper to enhance its scientific quality.

Dear reviewer, Many thanks for your detailed review and constructive comments/remarks for improvement. Much appreciated!

Introduction

First, I  would suggest that the authors delineate the knowledge gap in the literature and how they tend to contribute to narrowing such gap in the introduction. In lines, 58-64, the authors were trying to outline the organisation of the paper, which is ok but I would like them to change the word "chapters" to "sections".

As your suggestion, knowledge gap is clearly delineated now in the Abstract and Introduction sections. In addition the word ‘chapter’ is replaced with ‘section’ throughout the paper.

Methodology

In the methodology only the data collection tools were described but the data collected were analysed was not included. I suggest that the authors outline how the data was collected.

Methodology part is improved by explaining in details how the data was collected.

Results and discussion

I would suggest that the authors make section four results and discussion.

Section four is renamed as per your suggestion.

Round 2

Reviewer 1 Report

I find this version of the paper much improved as compared to the previous one. I still do no not find that the paper spells out clearly the knowledge gap that it wants to address, but the argument is much clearer now. I appreciate the care taken to add the necessary background information and keeping in mind that the readers may not be experts on Rwanda.  I really appreciate the quotes from fieldwork, which clearly show the difficulties and tensions people experienced in Rwanda around the land claims of returning refugees and displaced people. This really helps readers to gain an understanding of the realities of land sharing. In that regard, where in line 924/5 it says that land sharing supported peace, unity and reconciliation, I think it would be more in line with the evidence presented to state that land sharing policies intended to do that but did not - or only to a limited extent- achieve it.

My remaining observations relate to the section on the reconstruction period (4.4). The authors suggest here that the deficiencies and problems left by the earlier periods (emergency and early recovery) were basically 'fixed' (corrected) with the LTR policy between 2008-2014. The policy is presented as a success without much reservations, but I do not find that fully convincing. The reasons are the following: 1) the case cited in this section (949-50) is open ended, it is unclear how it was solved, what role the LTR played in that and whether those involved really felt the problem was settled. This case is open-ended, so how does it support the conclusion that the LTR was a success? 2) this section cites far less field evidence than the previous sections. Where quotes in previous sections make clear what tensions arose around land sharing, this section on the LTR does not on its turn cite people who were satisfied with how their cases were handled in this period. 3) the claim of the success of the LTR now is substantiated with reference to figures (less than 1 percent of the parcels still in conflict), but how confident can the authors be that 'registered' means 'problem solved'? Only one report on the LTR is cited.

I understand that the authors may see the Rwandan policy as a case that deserves to be more widely known and perhaps adopted and I agree that we need such examples to further construct  land policies for conflict-affected settings. However, the claim of its success might be stated with some more reservation in view of the evidence presented. Could the authors indicate whether other sources exist that confirm or perhaps contest this positive viewpoint?

Finally, some other points:

- What is the source of Table 2?

- line 1106- it is recommended that land issues should have been addressed earlier on. That may be easier said than done. How realistic would it have been to expect that, or how realistic would it be to expect this in any post-conflict setting?

- the authors claim lack of space. There was a basic explanation about land in lines 267-277 that in my opinion could be removed given the expected level of knowledge of the readership, which allow them to elaborate on some other elsewhere

- I see improvement in the English and I appreciate the efforts taken by the authors in this regard. However, the tekst still contains many sentences that are grammatically incorrect.

Author Response

Dear reviewer, Many thanks for your review and comments for improvement.

in line 417/8 it says that land sharing supported peace, unity and reconciliation - Improved with intended to support 

My remaining observations relate to the section on the reconstruction period (4.4). ...LTR... 

Section 4.4 is improved in regard of the three observations with: reorganizing the section 4.4 part, adding one new paragraph and appointing and explaining the sources. 

- What is the source of Table 2?

Secondary data collected from the fieldwork and used in the MSc research [3].

- I see improvement in the English and I appreciate the efforts taken by the authors in this regard. However, the tekst still contains many sentences that are grammatically incorrect.

English language has been improved and edited with assistance of experienced Native English speaking .

Reviewer 2 Report

Considering that the author(s) have been able to make all the necessary changes I requested them in my initial review. I am willing to accept this manuscript for publication in its current form, pending some very minor expressions (and language edits) that can be weeded off during copy editing.  

Author Response

pending some very minor expressions (and language edits) that can be weeded off during copy editing.  

Dear reviewer, Many thanks for your review and comments for improvement.

English language has been improved and edited with assistance of experienced Native English speaking person from our field of study.